# Streamlining segmentation of cryo-electron tomography datasets with Ais

**Mart GF Last[1]\*, Leoni Abendstein[1,2], Lenard M Voortman[1], Thomas H Sharp[1,3]\***

[1]Department of Cell and Chemical Biology, Leiden University Medical Center, Leiden, Netherlands; [2]Institute of Science and Technology Austria (ISTA), Klosterneuburg, Austria; [3]School of Biochemistry, University of Bristol, Bristol, United Kingdom

## eLife Assessment

This work describes a new software platform for machine-learning-based segmentation of and particle-picking in cryo-electron tomograms. The program and its corresponding online database of trained models will allow experimentalists to conveniently test different models and share their results with others. The paper provides **convincing** evidence that the software will be **valuable** to the community.

**\*For correspondence:**
mgflast@gmail.com (MGFL);
t.sharp@bristol.ac.uk (THS)

**Competing interest:** The authors declare that no competing interests exist.

**Abstract** Segmentation is a critical data processing step in many applications of cryo-electron tomography. Downstream analyses, such as subtomogram averaging, are often based on segmentation results, and are thus critically dependent on the availability of open-source software for accurate as well as high-throughput tomogram segmentation. There is a need for more user-friendly, flexible, and comprehensive segmentation software that offers an insightful overview of all steps involved in preparing automated segmentations. Here, we present Ais: a dedicated tomogram segmentation package that is geared towards both high performance and accessibility, available on GitHub. In this report, we demonstrate two common processing steps that can be greatly accelerated with Ais: particle picking for subtomogram averaging, and generating many-feature segmentations of cellular architecture based on in situ tomography data. Featuring comprehensive annotation, segmentation, and rendering functionality, as well as an open repository for trained models at aiscryoet.org, we hope that Ais will help accelerate research and dissemination of data involving cryoET.

## Introduction

Segmentation is a key step in processing cryo-electron tomography (cryoET) datasets that entails identifying specific structures of interest within the volumetric data and marking them as distinct features. This process forms the basis of many subsequent analysis steps, including particle picking for subtomogram averaging and the generation of 3D macromolecular feature maps that help visualize the ultrastructure of the sample.

Although it is a common task, the currently available software packages for tomogram segmentation often leave room for improvement in either scope, accessibility, or open availability of the source code. Some popular programs, such as Amira (Thermo Fisher Scientific), are not free to use, while other more general purpose EM data processing suites offer limited functionality in terms of visualization and user interaction. Specifically, we found that a deficit existed of software that is easy to use, competitively performing, freely available, and dedicated to segmentation of cryoET datasets for downstream processing.

In this report, we present Ais, an open-source tool that is designed to enable any cryoET user – whether experienced with software and segmentation or a novice – to quickly and accurately

segment their cryoET data in a streamlined and largely automated fashion. Ais comprises a comprehensive and accessible user interface within which all steps of segmentation can be performed, including: the annotation of tomograms and compiling datasets for the training of convolutional neural networks (CNNs), training and monitoring performance of CNNs for automated segmentation, 3D visualization of segmentations, and exporting particle coordinates or meshes for use in downstream processes. To help generate accurate segmentations, the software contains a library of various neural network architectures and implements a system of configurable interactions between different models. Overall, the software thus aims to enable a streamlined workflow where users can interactively test, improve, and employ CNNs for automated segmentation. To ensure compatibility with other popular cryoET data processing suites, Ais employs file formats that are common in the field, using .mrc files for volumes, tab-separated .txt or .star files for particle datasets, and the .obj file format for exporting 3D meshes.

To demonstrate the use of Ais, we outline its use in two such tasks: first, to automate the particle picking step of a subtomogram averaging workflow, and second for the generation of rich three-dimensional visualizations of cellular architecture, with ten distinct cellular components, based on cryoET datasets acquired on cellular samples.

## Results and discussion

The first step in image segmentation using CNNs is to manually annotate a subset of the data for use as a training dataset. Ais facilitates this step by providing a simple interface for browsing data, drawing overlays, and selecting boxes to use as training data (*Figure 1A*, *Figure 1—figure supplements 1–4*). Multiple features, such as membranes, microtubules, ribosomes, and mitochondrial granules, can be segmented and edited at the same time across multiple datasets (even hundreds). These annotations are then extracted and used as ground truth labels upon which to condition multiple neural networks, each trained to segment a single feature type, with which one can automatically segment the same or any other dataset (*Figure 1B*). Segmentation in Ais is performed *on-the-fly* and can achieve interactive framerates, depending on the size of the datasets and network architectures that are used. With a little experience, users can generate a training dataset and then train, apply, and asses the quality of a model within a few minutes (*Figure 1C*), including on desktop or laptop Windows and Linux systems with relatively low-end GPUs (e.g. we often use an NVIDIA T1000).

Many cryoET datasets look alike, especially for cellular samples. A model prepared by one user to segment, for example, ribosomes in a dataset with a pixel size of 10 Å, might also be adequate for another user's ribosome segmentation at 12 Å per pixel. To facilitate this sort of reuse and sharing of models, we launched an open model repository at aiscryoet.org where users can freely upload and download successfully trained models (*Figure 1D*). Models that pass screening become public, are labelled with relevant metadata (*Figure 1E*), and can be downloaded in a format that allows for direct use in Ais. Thus, users can skip the annotation and training steps of the segmentation workflow. To kickstart the repository, all 27 models that are presented in this article have been uploaded to it.

Our software is not the first to address the challenge of segmenting cryoET datasets; established suites such as EMAN2 (*Galaz-Montoya et al., 2015*), MIB (*Belevich et al., 2016*), SuRVoS (*Luengo et al., 2017*), or QuPath (*Bankhead et al., 2017*) also provide some or most of the functionality that is available in Ais. Each comes equipped with one or various choices of neural network architectures to use, and many more designs for neural networks for semantic image segmentation can be found in the literature. Therefore, as well as creating a package geared specifically towards ease of use and fast results, we also wanted to include functionality that enables a user to quickly compare different models in order to facilitate determining which models are best suited for a particular segmentation task. The software thus includes a library of a number of well-performing architectures, including adaptations of single-model CNN architectures such as InceptionNet (*Szegedy et al., 2014*), ResNet (*Szegedy et al., 2016*), various UNets (*Ronneberger et al., 2015*), VGGNet (*Simonyan and Zisserman, 2015*), and the default architecture available in EMAN2 *Galaz-Montoya et al., 2015*, as well as the more complex generative-adversarial network Pix2pix (*Isola et al., 2017*). This library can also be extended by copying any Python file that adheres to a minimal template into the corresponding directory of the project (Appendix 1).

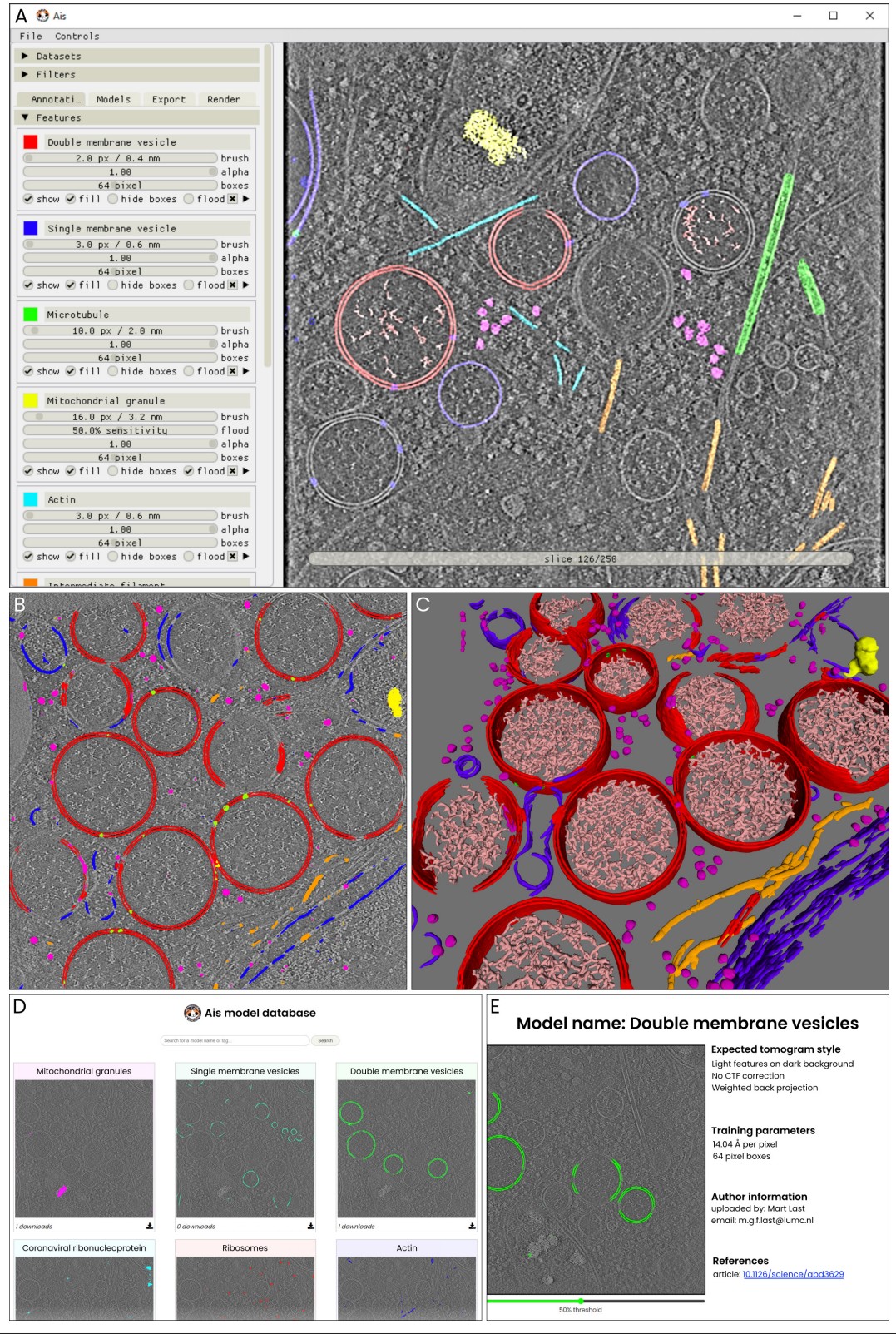

**Figure 1.** An overview of the user interface and functionalities. The various panels represent sequential stages in the Ais processing workflow, including annotation (**A**), testing convolutional neural networks (CNNs) (**B**), and visualizing segmentation (**C**). These images (**A–C**) are unedited screenshots of the software. (**A**) The interface for annotation of datasets. In this example, a tomographic slice has been annotated with various features – a detailed explanation follows in *Figure 5*. (**B**) After annotation, multiple neural networks are set up and trained on the aforementioned annotations. The resulting

*Figure 1 continued on next page*

*Figure 1 continued*

models can then be used to segment the various distinct features. In this example, double-membrane vesicles (double membrane vesicles DMVs, red), single membranes (blue), ribosomes (magenta), intermediate filaments (orange), mitochondrial granules (yellow), and molecular pores in the DMVs (lime) are segmented. (**C**) After training or downloading the required models and exporting segmented volumes, the resulting segmentations are immediately available within the software for 3d rendering and inspection. (**D**) The repository at aiscryoet.org facilitates the sharing and reuse of trained models. After validation, submitted models can be freely downloaded by anyone. (**E**) Additional information, such as the pixel size and the filtering applied to the training data, is displayed alongside all entries in the repository, in order to help a user identify whether a model is suited to segment their datasets.

The online version of this article includes the following figure supplement(s) for figure 1:

**Figure supplement 1.** Ais annotation interface.

**Figure supplement 2.** Ais training and model testing interface.

**Figure supplement 3.** Ais volume exporting interface.

**Figure supplement 4.** Ais 3D rendering interface.

## A library of neural network architectures supports varied applications

To illustrate how useful it can be to rapidly test various architectures before selecting one that is well suited for the segmentation of any particular feature, we used six different architectures for the segmentation of three distinct features within the same tomogram, and analyzed the results (*Table 1*). We used a cryoET dataset that we had previously acquired (*Abendstein et al., 2023*), which contained liposomes with membrane-bound Immunoglobulin G3 (IgG3) antibodies that form an elevated Fragment crystallizable (Fc) platform, prepared on a lacey carbon substrate. The features of interest for segmentation were the membranes, antibody platforms, and carbon support film (*Figure 2A*).

After training, we applied the resulting models to a different tomogram containing the same features, that were not previously used to generate the training data (*Figure 2B*), so that there was no overlap between the training and testing datasets. Next, we compared the training times, relative loss values, and quality of the segmentations.

Based on the loss values (the loss is a metric of how well predictions match the ground truth labels), VGGNet was best suited for the segmentation of membranes, while UNet performed best on the carbon support film and antibody platforms. However, the loss values do not capture model performance in the same way as human judgement (*Figure 2C*). For the antibody platform, the model that would be expected to be one of the worst based on the loss values, Pix2pix, actually generates segmentations that are well-suited for the downstream processing tasks. It also outputs fewer false positive segmentations for sections of membranes than many other models, including the lowest-loss model UNet. Moreover, since Pix2pix is a relatively large network, it might also be improved further

**Table 1.** Comparison of some of the default models available in Ais.

[a]The computational cost is only roughly proportional to the number of model parameters, which is reported in the software. The specifics of the network architecture affect the processing speed more significantly. [b]Time required to process one 511 × 720 pixel-sized tomographic slice. [c]These columns list the loss values after training, calculated as the binary cross-entropy (bce) between the prediction and original annotation. The loss is a (rough) metric of how well a trained network performs (see Methods). [d]Unlike the other architectures, Pix2pix is not trained to minimize the bce loss but uses a different loss function instead. The bce loss values shown here were computed after training and may not be entirely comparable.

| Architecture | Parameters | Training time[a] | Processing time[a, b] | Membrane[c] | Carbon[c] | Antibody[c] |
|---|---|---|---|---|---|---|
| EMAN2 | 378,081 | 38 s | 50 ms | 0.044 | .072 | .010 |
| InceptionNet | 550,529 | 462 s | 295 ms | 0.046 | .120 | .008 |
| UNet | 922,881 | 132 s | 110 ms | 0.013 | **.007** | **.001** |
| VGGNet | 1,493,059 | 91 s | 85 ms | **0.011** | .125 | .013 |
| ResNet | 4,887,265 | 1533 s | 980 ms | 0.047 | .060 | .014 |
| Pix2pix[d] | 29,249,409 | 793 s | 225 ms | 0.050[d] | .129[d] | .016[d] |

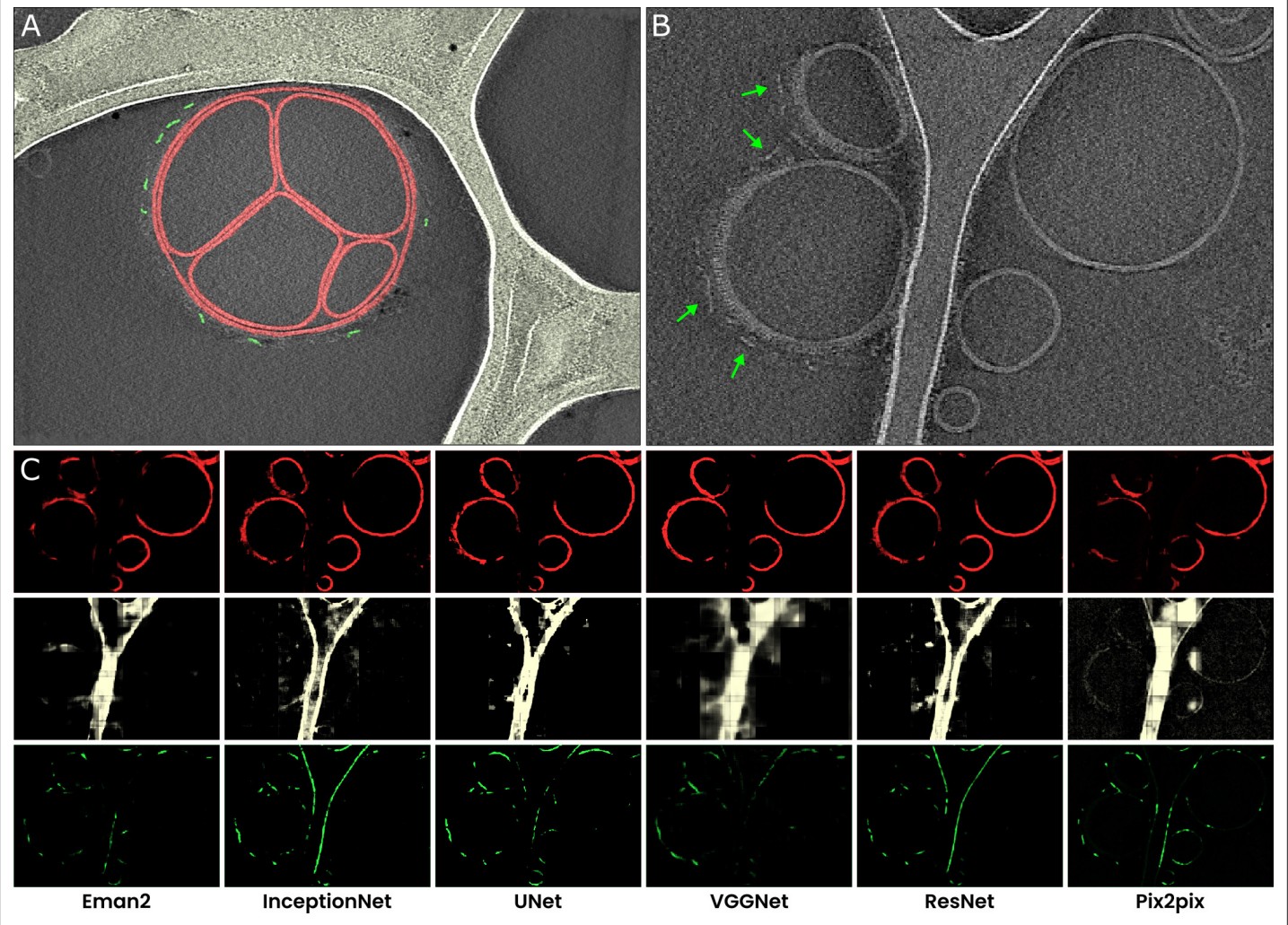

**Figure 2.** A comparison of different neural networks for tomogram segmentation. (**A**) A representative example of the manual segmentation used to prepare training datasets. Membranes are annotated in red, carbon film in bright white, and antibody platforms in green. For the antibody training set, we used annotations prepared in multiple slices of the same tomogram, but for the carbon and membrane training set the slice shown here comprised all the training data. (**B**) A tomographic slice from a different tomogram that contains the same features of interest, also showing membrane-bound antibodies with elevated Fc platforms that are adjacent to carbon (red arrowheads). (**C**) Results of segmentation of membranes (top; red), carbon (middle; white), and antibody platforms (bottom; green), with the six different neural networks.

The online version of this article includes the following figure supplement(s) for figure 2:

**Figure supplement 1.** Comparison of manual annotations and UNet, Pix2pix, and improved Pix2pix antibody platform segmentations for the data in *Figure 2*.

by increasing the number of training epochs. We thus decided to use Pix2pix for the segmentation of antibody platforms, and increased the size of the antibody platform training dataset (from 58 to 170 positive samples) to train a much improved second iteration of the network for use in the following analyses (*Figure 2—figure supplement 1*).

When taking the training and processing speeds into account as well as the segmentation results, there is no overall best architecture. We, therefore, included multiple well-performing model architectures in the final library, in order to allow users to select from these models to find one that works well for their specific datasets. Although it is not necessary to screen different network architectures and users may simply opt to use the default (VGGNet), these results thus show that it can be useful to test different networks to identify one that is best. Moreover, these results also highlight the utility of preparing well-performing models by iteratively improving training datasets and re-training models in a streamlined interface. To aid in this process, the software displays the loss value of a network during

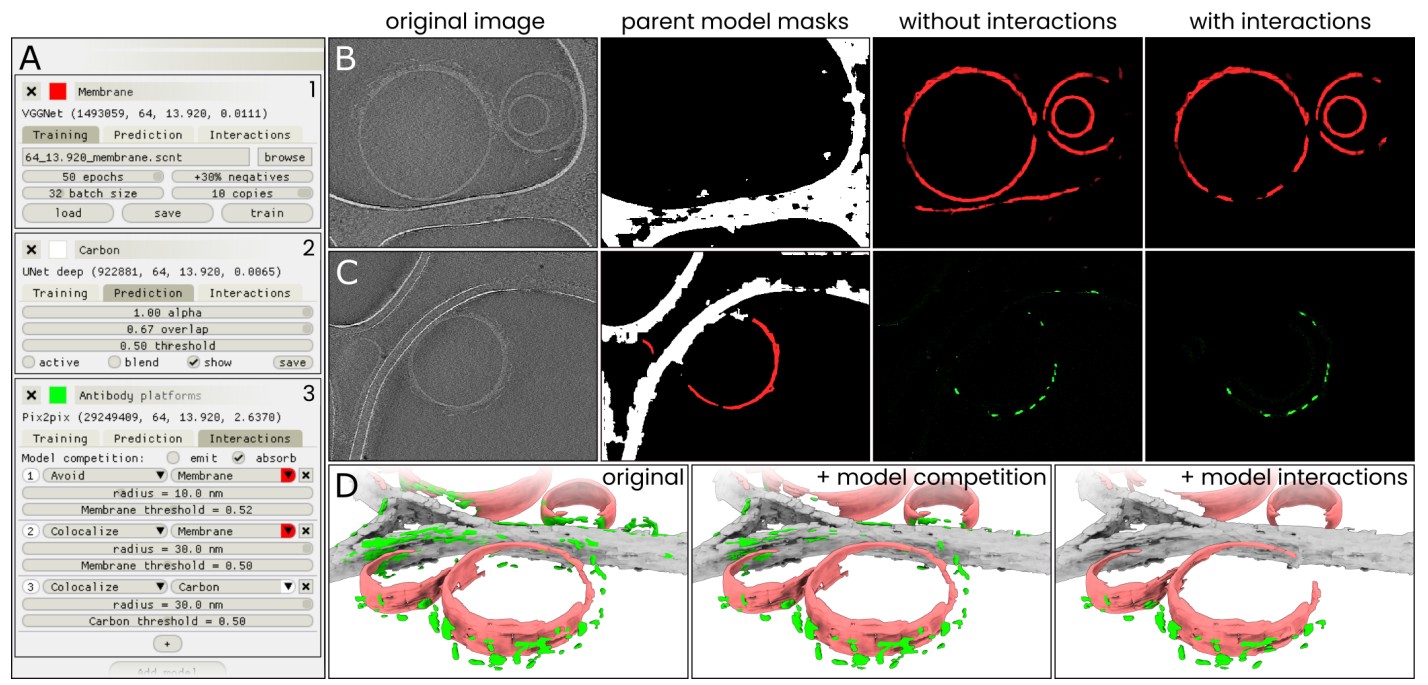

**Figure 3.** Model interactions can significantly increase segmentation accuracy. (**A**) An overview of the settings available in the 'Models' menu in Ais. Three models: (1) 'membrane' (red), (2) 'carbon' (white), and (3) 'antibody platforms' (green) are active, with each showing a different section of the model settings: the training menu (1), prediction parameters (2), and the interactions menu (3). (**B**) A section of a tomographic slice is segmented by two models, carbon (white; *parent* model) and membrane (red; *child* model), with the membrane model showing a clear false positive prediction on an edge of the carbon film (panel 'without interactions'). By configuring an avoidance interaction between the membrane model that is conditional upon the carbon model's prediction, this false positive is avoided (panel 'with interactions'). (**C**) By setting up multiple model interactions, inaccurate predictions by the 'antibody platforms' model are suppressed. In this example, the membrane model avoids carbon while the antibody model is set to colocalize with the membrane model. (**D**) 3D renders (see Methods) of the same dataset as used in *Figure 2* processed three ways: without any interactions (left), using model competition only (middle), or by using model competition as well as multiple model interactions (right).

training and allows for the application of models to datasets during training. Thus, users can inspect how a model's output changes during training and decide whether to interrupt training and improve the training data or choose a different architecture.

## Fine-tuning segmentation results with *model interactions*

Although the above results go some way towards distinguishing the three different structures, they also show demonstrate a common limitation encountered in automated tomogram segmentation: some parts of the image are assigned a high segmentation value by multiple of the networks, leading to false classifications and ambiguity in the results. For example, the InceptionNet and ResNet antibody platform models falsely label the edges of the carbon film.

To further improve the segmentation results, we decided to implement a system of proximity-based 'model interactions' of two types, colocalization and avoidance (*Figure 3A*), using which the output of one model can be adjusted based on the output of other models. In a colocalization interaction, the predictions of one model (the *child*) are suppressed wherever the prediction value of another model (the *parent*) is below some threshold. In an avoidance interaction, suppression occurs wherever the parent model's prediction value is above a threshold.

These interactions are implemented as follows: first, a binary mask is generated by thresholding the *parent* model's predictions using a user-specified threshold value. Next, the mask is then dilated using a circular kernel with a radius $R$, a parameter that we call the interaction radius. Finally, the *child* model's prediction values are multiplied with this mask.

Besides these specific interactions between two models, the software also enables pitching multiple models against one another in what we call 'model competition'. Models can be set to 'emit' and/or 'absorb' competition from other models. Here, to emit competition means that a model's prediction

value is included in a list of competing models. To absorb competition means that a model's prediction value will be compared to all values in that list, and that this model's prediction value for any pixel will be set to zero if any of the competing models' prediction value is higher. On a pixel-by-pixel basis, all models that absorb competition are thus suppressed whenever their prediction value for a pixel is lower than that of any of the emitting models.

With the help of these model interactions, it is possible to suppress common erroneous segmentation results. For example, an interaction like 'absorbing membrane model avoids emitting carbon model with R = 10 nm' is effective at suppressing the prediction of edges of the carbon film as being membranes (*Figure 3B*). Another straightforward example of the utility of membrane interactions is the segmentation of membrane-bound particles. By defining the following two interactions: 'antibody platform model avoids membrane model with R = 10 nm' followed by 'antibody platform model colocalizes with membrane model with R = 30 nm', the Fc-platforms formed by IgG3 at a distance of ~22 nm from the membrane are retained, while false positive labelling of features such as the membrane or carbon is suppressed (*Figure 3C*).

By conditionally combining and editing the prediction results of multiple neural networks, model interactions can thus be helpful in fine-tuning segmentations to be better suited for downstream applications. To illustrate this, we generated a comparison of segmentation results using (i) no interactions, (ii) model competition only, and (iii) model competition as well as model interactions (*Figure 3D*), which demonstrates the degree to which false positives can be reduced by the use of model interactions (although at times at the expense of increasing the rate of false negatives).

## Automating particle picking for subtomogram averaging

Protein structure determination by cryoET requires the careful selection of many subtomograms (i.e. sub-volumes of a tomogram that all contain the same structure of interest), and aligning and averaging these to generate a 3D reprojection of the structure of interest with a significantly increased signal to noise ratio. The process of selecting these sub-volumes is called 'particle picking,' and can be done either manually or in an automated fashion. Much time can be saved by automating particle picking based on segmentations. In many cases, though, segmentation results are not readily usable for particle picking, as they can often introduce numerous false positives. This is particularly the case with complex, feature-rich datasets such as those obtained within cells, where the structures of interest can visually appear highly similar to other structures that are also found in the data, or when the structures of interest are located close to other features and are, therefore, hard to isolate. An example of this latter case is the challenge of picking membrane-bound particles.

Recently, we have used cryoET and subtomogram averaging to determine the structures of membrane-bound IgG3 platforms and of IgG3 interacting with the human complement system component 1 (C1) on the surface of lipid vesicles (*Abendstein et al., 2023*). The reconstructions of the antibody platforms alone and of the antibody-C1 complex were prepared using 1193 and 2561 manually selected subtomograms, extracted from 55 and 101 tomograms, respectively. Manual picking of structures of interest, although very precise, is very time-consuming and in this particular case took approximately 20 hr of work.

To demonstrate the utility of our software for particle picking, we re-analyzed these same datasets, this time using Ais to automate the picking of the two structures: antibody platforms and antibody-C1 complexes. For the antibody platforms, we used the same models and model interactions as described above, while we trained an additional neural network to identify C1 complexes for the segmentation of the antibody-C1 complexes. To prepare a training dataset for this latter model, we opened all 101 tomograms in Ais and browsed the data to select and annotate slices where one or multiple antibody-C1 complexes were clearly visible (*Figure 4A*). The training dataset thus consisted of samples taken from multiple different tomograms; the annotation and data selection in this case took around 1 hr of work.

The antibody platform and antibody-C1 complex models were then applied to the respective datasets, in combination with the membrane and carbon models and the model interactions described above (*Figure 4B*): the membrane avoiding carbon, and the antibody platforms colocalizing with the resulting membranes. We then launched a relatively large batch segmentation process (3 models × 156 tomograms) and left it to complete overnight.

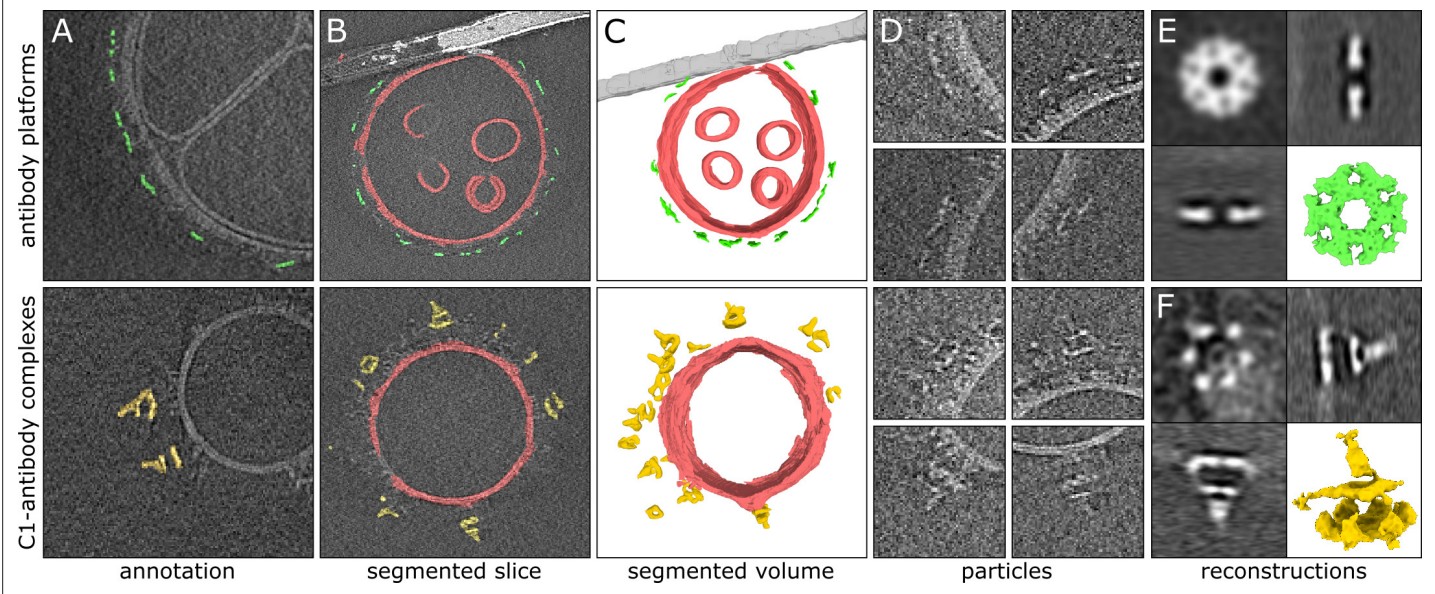

**Figure 4.** Automated particle picking for sub-tomogram averaging of antibody complexes. (**A**) Manually prepared annotations used to train a neural network to recognize antibody platforms (top) or antibody-C1 complexes (bottom). (**B**) Segmentation results as visualized within the software. Membranes (red) and carbon support film (white) were used to condition the antibody (green) and antibody-C1 complex (yellow) predictions using model interactions. (**C**) 3D representations of the segmented volumes rendered in Ais. (**D**) Tomographic slices showing particles picked automatically based on the segmented volume shown in panel c. (**E**) Subtomogram averaging result of the 2499 automatically picked antibody platforms. (**F**) Subtomogram averaging result obtained with the 602 automatically picked antibody-C1 complexes. The quadrants in panels e and f show orthogonal slices of the reconstructed density maps and a 3D isosurface model (the latter rendered in ChimeraX [*Goddard et al., 2018*]).

The online version of this article includes the following figure supplement(s) for figure 4:

**Figure supplement 1.** A visual (2D) representation of the processing steps employed in the automated picking process.

**Figure supplement 2.** Inspecting the autopicking results in the Ais renderer.

**Figure supplement 3.** Comparison of the automatically picked C1-IgG3 complex reconstruction versus the original reconstruction in *Abendstein et al., 2023*.

Once complete, we used Ais to inspect the segmented volumes and original datasets in 3D, and adjusted the threshold value so that, in as far as possible, only the particles of interest remained visible and the number of false positive particles was minimized (*Figure 4C*). After selecting these values, we then launched a batch particle-picking process to determine lists of particle coordinates based on the segmented volumes.

Particle picking in Ais comprises a number of processing steps (*Figure 4*, *Figure 4—figure supplement 1*). First, the segmented (.mrc) volumes are thresholded at a user-specified level. Second, a distance transform of the resulting binary volume is computed, in which every nonzero pixel in the binary volume is assigned a new value, equal to the distance of that pixel to the nearest zero-valued pixel in the mask. Third, a watershed transform is applied to the resulting volume, so that the sets of pixels closest to any local maximum in the distance transformed volume are assigned to one group. Fourth, groups that are smaller than a user-specified minimum volume are discarded. Fifth, the remaining groups are assigned a weight value, equal to the sum of the prediction value (i.e. the corresponding pixel value in the input.mrc volume) of the pixels in the group. For every group found within close proximity to another group (using a user-specified value for the minimum particle spacing), the group with the lower weight value is discarded. Finally, the centroid coordinate of the grouped pixels is considered the final particle coordinate, and the list of all coordinates is saved in a tab-separated text file with values rounded to the nearest integer.

As an alternative output format, segmentations can also be converted to and saved as triangulated meshes (.obj file format), which can then be used for, e.g., membrane-guided particle picking (*Pyle et al., 2022*). After picking particles, the resulting coordinates are immediately available for inspection in the Ais 3D renderer (*Figure 4*, *Figure 4—figure supplement 2*).

After picking, we used EMAN2 (*Galaz-Montoya et al., 2015*, spt_boxer.py) to extract volumes using the particle coordinates as an input (*Figure 4D*), which resulted in 2499 vol for the antibody platform reconstruction and 602 for the antibody-C1 complex (*n.b.* these numbers can be highly dependent on the threshold value). These volumes, or subtomograms, were used as the input for subtomogram averaging using EMAN2 *Galaz-Montoya et al., 2015* and Dynamo (*Castaño-Díez et al., 2012*; see Materials and methods), without further curation – i.e., we did not manually discard any of the extracted volumes.

After applying the same approach to subtomogram averaging as used previously (*Abendstein et al., 2023*), the resulting averages were indeed highly similar to the original reconstructions (*Figure 4E and F*, *Figure 4—figure supplement 3*). These results demonstrate that Ais can be successfully used to automate particle picking, and thus to significantly reduce the amount of time spent on what is often a laborious processing step.

## Many-feature segmentations of complex in situ datasets

Aside from particle picking, segmentation is also often used to visualize and study the complex internal structure of a sample, as for example encountered when applying tomography to whole cells. Here too, the accuracy of a segmentation can be a critical factor in the success of downstream analyses such as performing measurements on the basis of 3D feature maps generated *via* segmentation.

A challenging aspect of the segmentation of cellular samples is that these datasets typically contain many features that are biologically distinct, but visually and computationally difficult to distinguish. For example, one challenge that is often encountered is that of distinguishing between various linearly shaped components: lipid membranes, actin filaments, microtubules, and intermediate filaments, which all appear as linear features with a relatively high density. To show the utility of Ais for the accurate segmentation of complex cellular tomograms, we next demonstrate a number of examples of such feature-rich segmentations.

The first example is a segmentation of seven distinct features observed in the base of *Chlamydomonas reinhardtii* cilia (*Figure 5A*), using the data by *van den Hoek et al., 2022* that was deposited in the Electron Microscopy Public Image Archive (EMPIAR, *Iudin et al., 2023*) with accession number 11078. The features are: membranes, ribosomes, microtubule doublets, axial microtubules, non-microtubular filaments, interflagellar transport trains (IFTs), and glycocalyx. This dataset was particularly intricate (the supplementary information to the original publication lists more than 20 features that can be identified across the dataset) and some rare features, such as the IFTs, required careful annotation across all tomograms before we could compile a sufficiently large training dataset. The final segmentation correctly annotates most of the selected characteristics present in the sample: the ribosome exclusion zone that surrounds the ciliary base (*van den Hoek et al., 2022*) is clearly recognizable, and the structures of the glycocalyx, membranes, and microtubule doublets within the cilia are well defined. Some fractions of the meshwork of stellate fiber and Y-link proteins are also detected within the cilium.

In the second example, we show a segmentation of six features found in and around a mitochondrion in a mouse neuron (*Figure 5B*), using the data by *Wu et al., 2023* as available in the Electron Microscopy Data Bank (EMDB) (*Lawson et al., 2016*) with accession number 29207. In the original publication, the authors developed a segmentation method to detect and perform measurements on the granules found within mitochondria. Using Ais, we were able to prepare models to segment these granules, as well as microtubules, actin filaments, and ribosomes, and to distinguish between the highly similar vesicular membranes on the one hand, and the membranes of the mitochondrial cristae on the other.

Lastly, we used Ais to generate a 3D visualization of ten distinct cellular features observed in coronavirus-infected mammalian cells (*Figure 5C*, *Figure 5—figure supplement 1*), using data by *Wolff et al., 2020*: single membranes, double-membrane vesicles (DMVs), actin filaments, intermediate filaments, microtubules, mitochondrial granules, ribosomes, coronaviral nucleocapsid proteins, coronaviral pores in the DMVs, and the nucleic acids within the DMV replication organelles. The software was able to accurately distinguish between single membranes and double membranes, as well as to discriminate between the various filaments of the cytoskeleton. Moreover, we could identify the molecular pores within the DMV, and pick sets of particles that might be suitable for use in subtomogram averaging (see *Figure 5*, *Figure 5—figure supplement 2*). We could also segment the

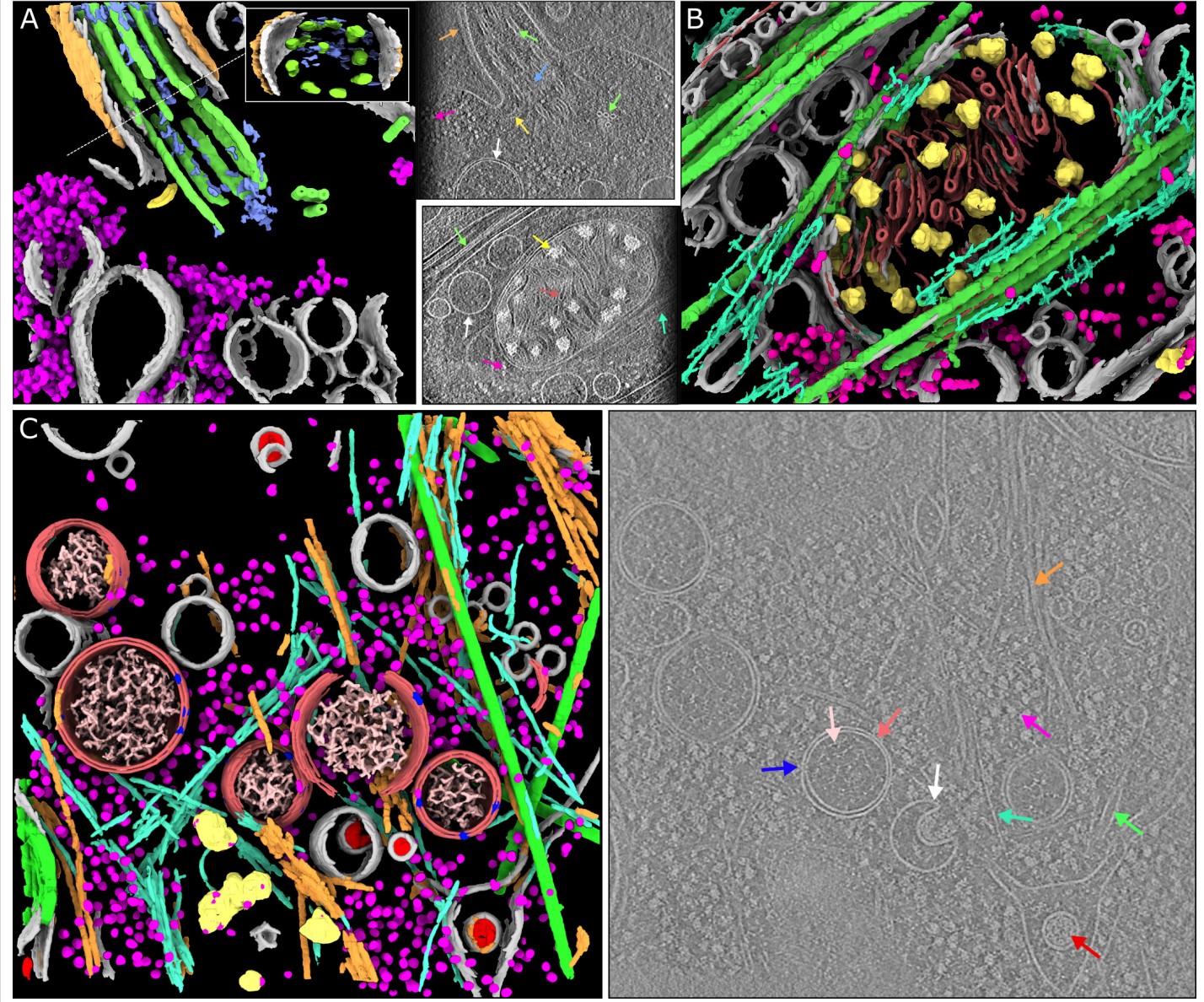

**Figure 5.** Segmentations of complex in situ tomograms. (**A**) A segmentation of seven distinct features observed in the base of *C. reinhardtii* cilia (***van den Hoek et al., 2022***, EMPIAR-11078, tomogram 12): membranes (gray), ribosomes (magenta), microtubule doublets (green) and axial microtubules (green), non-microtubular filaments within the cilium (blue), inter-flagellar transport trains (yellow), and glycocalyx (orange). Inset: a perpendicular view of the axis of the cilium. The arrows in the adjacent panel indicate these structures in a tomographic slice. (**B**) A segmentation of six features observed in and around mitochondria in a mouse neuron with Huntington's disease phenotype (***Wu et al., 2023***) (EMD-29207): membranes (gray), mitochondrial granules (yellow), membranes of the mitochondrial cristae (red), microtubules (green), actin (turquoise), and ribosomes (magenta). (**C**) Left: a segmentation of ten different cellular components found in tomograms of coronavirus-infected mammalian cells (***Wolff et al., 2020***): double-membrane vesicles (double membrane vesicles DMVs, light red), single membranes (gray), viral nucleocapsid proteins (red), viral pores in the DMVs (blue), nucleic acids in the DMVs (pink), microtubules (green), actin (cyan), intermediate filaments (orange), ribosomes (magenta), and mitochondrial granules (yellow). Right: a representative slice, with examples of each of the features (except the mitochondrial granules) indicated by arrows of the corresponding colour.

The online version of this article includes the following figure supplement(s) for figure 5:

**Figure supplement 1.** Individual images of the ten features segmented in the coronavirus-infected mammalian cells dataset (***Wolff et al., 2020***, Science).

**Figure supplement 2.** A collage of automatically picked molecular pores in the double membrane vesicles.

nucleocapsid proteins, thus distinguishing viral particles from other, similarly sized, single membrane vesicles, as well as detect the nucleic acids found within the DMVs. Although manual annotation took some hours in this case, the processing of the full volume took approximately 3 min per feature once the models were trained, and models can of course be applied to other volumes without requiring additional training.

To conclude, our aim with the development of Ais was to simplify and improve the accuracy of automated tomogram segmentation in order to make this processing step more accessible to many cryoET users. Here, we have attempted to create an intuitive and organized user interface that stream-lines the whole workflow from annotation, to model preparation, to volume processing and particle picking and inspecting the results. Additionally, the model repository at aiscryoet.org is designed to aid users in achieving results even faster, by removing the need to generate custom models for common segmentation tasks. To help users become familiar with the software, documentation and tutorials are available at ais-cryoet.readthedocs.org and video tutorials can be accessed via youtube.com/@scNodes. By demonstrating the use of Ais in automating segmentation and particle picking for subtomogram averaging, and making the software available as an open-source project, we thus hope to help accelerate research and dissemination of data involving cryoET.

# Materials and methods
## Neural network comparisons

The comparison presented in *Table 1* was prepared with the use of the same training datasets for all architectures, consisting of 53/52/58 positive images showing membranes/carbon/antibody plat-forms and corresponding annotations alongside 159/51/172 negative images that did not contain the feature of interest, but rather the other two features, reconstruction artefacts, isolated protein, or background noise. Images were 64 × 64 pixels in size and the dataset was resampled, in random orientations, such that every positive image was copied 10 times and that the ratio of negatives to positives was 1.3:1. Networks were trained for 50 epochs with 32 images per batch, with the excep-tion of the ResNet and Pix2pix antibody platform models, which were trained for 30 epochs to avoid a divergence that occurred during training after a larger number of epochs, due to the large number of network parameters and relatively low number of unique input images.

The reported loss is that calculated on the training dataset itself, i.e., no validation split was applied. During regular use of the software, users can specify whether to use a validation split or not. By default, a validation split is not applied in order to prioritize training accuracy by making full use of the input set of ground truth annotations. When the training dataset is sufficiently large, we recommend increasing the size of the validation split. Depending on the chosen split size, the software reports either the overall training loss or the validation loss during training.

## Data visualization

Images shown in the figures were either captured within the software (*Figures 1 and 3BC* 'orig-inal image', *Figure 4ABC*), output from the software that was colourized in Inkscape (*Figures 2C and 3BC*), or output from the software that was rendered using ChimeraX[11] (*Figures 3D and 4EF*, *Figure 5*). For the panels in *Figure 3D*, segmented volumes were rendered as isosurfaces at a manu-ally chosen suitable isosurface level and with the use of the 'hide dust' function (the same settings were used for each panel, different settings used for each feature). This 'dust' corresponds to small (in comparison to the segmented structures of interest) volumes of false positive segmentations, which are present in the data due to imperfections in the used models. The rate and volume of false positives can be reduced either by improving the models (typically by including more examples of the images that would be false negatives or positives in the training data) or, if the dust particles are indeed smaller than the structures of interest, they can simply be discarded by filtering particles based on their volume, as applied here. In particle picking a 'minimum particle volume' is specified – particles with a smaller volume are considered 'dust'.

## Hardware

The software does not require a GPU, but works optimally when a CUDA-capable GPU is available. For the measurements shown in *Table 1* we used an NVIDIA Quadro P2200 GPU on a PC with an Intel

i9-10900K CPU. We've also extensively used the software on a less powerful system equipped with an NVIDIA T1000 and an Intel i3-10100 CPU, as well as on various systems with intermediate specifications, and found that the software reaches interactive segmentation rates in most cases. For batch processing of many volumes, a more powerful GPU is useful.

## Tomogram reconstruction and subtomogram averaging

Data collection and subtomogram averaging (*Figures 3 and 4*) was performed as described in a previously published article (*Abendstein et al., 2023*). Briefly, tilt series were collected on a Talos Arctica 200 kV system equipped with a Gatan K3 detector with an energy filter at a pixel size of 1.74 Å per pixel using a dose-symmetric tilt scheme with a range ±57° and tilt increments of 3° with a total dose of 60 e/Å (*Belevich et al., 2016*). Tomograms were reconstructed using IMOD (*Mastronarde and Held, 2017*). Particle picking was done in Ais (and is explained in more detail in the online documentation). Subtomogram averaging was done using a combination of EMAN (*Galaz-Montoya et al., 2015*) and Dynamo (*Castaño-Díez et al., 2012*). For a detailed description of the subtomogram averaging procedure, see *Abendstein et al., 2023*.

## Open-source software

This project depends critically on a number of open-source software components, including: Python, Tensorflow (*Abadi, 2015*), numpy (*Harris et al., 2020*), scipy (*Virtanen et al., 2020*), sci-kit-image (*van der Walt et al., 2014*), mrcfile (*Burnley et al., 2017*), and imgui (*Cornut, 2023*).

## Software availability

A standalone version of the software is available as 'Ais-cryoET' on the Python package index and on GitHub, (copy archived at *Last, 2024*), under the GNU GPLv3 license. We have also integrated the functionality into scNodes (*Last et al., 2023*), our dedicated processing suite for correlated light and electron microscopy. In the combined package, the segmentation editor contains additional features for visualization of fluorescence data and the scNodes correlation editor can be used to prepare correlated datasets for segmentation. Documentation for both versions of Ais can be found at ais-cryoet.readthedocs.org. Video tutorials are available via youtube.com/@scNodes.

## Acknowledgements

We thank A Koster and M Barcena for helpful discussions and kindly sharing the coronaviral replication organelle datasets. We are also grateful to *van den Hoek et al., 2022* and *Wu et al., 2023*, for uploading the data that we used for *Figure 5* onto EMPIAR and EMDB, as well as to the authors of various other datasets uploaded to these databases that are not discussed in this manuscript but that were useful for testing the software. We also thank the reviewers, whose comments were very helpful in improving the manuscript and the software. Finally, we are grateful the early Ais users who provided us with feedback on the software and reported issues. This research was supported by the following grants to THS: European Research Council H202 Grant 759517; European Union's Horizon Europe Program IMAGINE grant 101094250, and the Netherlands Organization for Scientific Research Grant VI.Vidi.193.014.

## Additional information

### Funding

| Funder | Grant reference number | Author |
| --- | --- | --- |
| European Research Council | H202 Grant 759517 | Thomas H Sharp |
| HORIZON EUROPE European Research Council | 10.3030/101094250 | Thomas H Sharp |

| Funder | Grant reference number | Author |
|---|---|---|
| Nederlandse Organisatie voor Wetenschappelijk Onderzoek | VI.Vidi.193.014 | Thomas H Sharp |

The funders had no role in study design, data collection and interpretation, or the decision to submit the work for publication.

## Author contributions

Mart GF Last, Conceptualization, Data curation, Software, Formal analysis, Validation, Investigation, Visualization, Methodology, Writing – original draft, Writing – review and editing; Leoni Abendstein, Resources, Data curation, Formal analysis, Validation, Investigation, Visualization, Methodology, Writing – original draft, Writing – review and editing; Lenard M Voortman, Supervision, Validation, Writing – original draft, Writing – review and editing; Thomas H Sharp, Supervision, Funding acquisition, Validation, Writing – original draft, Project administration, Writing – review and editing

## Author ORCIDs

Mart GF Last ⓘ https://orcid.org/0000-0002-3739-8863
Leoni Abendstein ⓘ https://orcid.org/0000-0001-7634-5353
Lenard M Voortman ⓘ https://orcid.org/0000-0001-9794-067X
Thomas H Sharp ⓘ https://orcid.org/0000-0002-1990-2333

Reviewer #1 (Public review): https://doi.org/10.7554/eLife.98552.3.sa1
Reviewer #2 (Public review): https://doi.org/10.7554/eLife.98552.3.sa2
Reviewer #3 (Public review): https://doi.org/10.7554/eLife.98552.3.sa3
Author response https://doi.org/10.7554/eLife.98552.3.sa4

# Additional files

## Supplementary files

- MDAR checklist

## Data availability

Source code is available via GitHub, (copy archived at *Last, 2024*), software available on the python package index as 'Ais-cryoET'. Trained neural networks have been uploaded to aiscryoet.org.

The following previously published datasets were used:

| Author(s) | Year | Dataset title | Dataset URL | Database and Identifier |
|---|---|---|---|---|
| Wu et al. | 2023 | CryoET tomogram of mitochondria in BACHD mouse model neuron | https://www.ebi.ac.uk/emdb/EMD-29207 | Massive, EMDB-29207 |
| Hoek V | 2022 | In situ cryo-electron tomography of the *C. reinhardtii* ciliary transition zone | https://www.ebi.ac.uk/empiar/EMPIAR-11078/ | Massive, EMPIAR-11078 |

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

## Appendix 1

## Supplementary note 1: Python file format for Ais models

Adding a Keras model

Most models in the standard Ais library are Keras models (tensorflow.keras). Adding an extra keras model with a new architecture is relatively straightforward and can be achieved by adding a.py file to Ais/models directory. The.py file requires three components: a title for the model, a boolean that specifies whether the model should be available in the software, and a function 'create' that returns a keras model. The implementation of the VGGNet model (vggnet.py) is copied below as an example.

```python
from tensorflow.keras.models import Model
from tensorflow.keras.layers import Input, Conv2D, MaxPooling2D,
Conv2DTranspose
from tensorflow.keras.optimizers import Adam

title = "VGGNet"
include = True

def create(input_shape):
inputs = Input(input_shape)

# Block 1
conv1 = Conv2D(64, (3, 3), activation='relu', padding='same')(inputs)
conv2 = Conv2D(64, (3, 3), activation='relu', padding='same')(conv1)
pool1 = MaxPooling2D(pool_size=(2, 2))(conv2)

# Block 2
conv3 = Conv2D(128, (3, 3), activation='relu', padding='same')(pool1)
conv4 = Conv2D(128, (3, 3), activation='relu', padding='same')(conv3)
pool2 = MaxPooling2D(pool_size=(2, 2))(conv4)

# Block 3
conv5 = Conv2D(256, (3, 3), activation='relu', padding='same')(pool2)
conv6 = Conv2D(256, (3, 3), activation='relu', padding='same')(conv5)
pool3 = MaxPooling2D(pool_size=(2, 2))(conv6)

# Upsampling and Decoding
up1 = Conv2DTranspose(128, (2, 2), strides=(2, 2), padding='same')(pool3)
conv7 = Conv2D(128, (3, 3), activation='relu', padding='same')(up1)

up2 = Conv2DTranspose(64, (2, 2), strides=(2, 2), padding='same')(conv7)
conv8 = Conv2D(64, (3, 3), activation='relu', padding='same')(up2)

up3 = Conv2DTranspose(1, (2, 2), strides=(2, 2), padding='same')(conv8)
output = Conv2D(1, (1, 1), activation='sigmoid')(up3)

# create the model
model = Model(inputs=[inputs], outputs=[output])
model.compile(optimizer=Adam(), loss='binary_crossentropy')

return model
```

## Adding a custom model

Adding a non-Keras model is also possible but requires a little bit of extra work. Only a small number of methods of the Keras model object type are directly accessed by Ais. These are: count_params, fit, predict, save, and load. Adding a custom model thus requires adding a.py file to the *Ais/models* that contains four components: a title, a boolean that specifies whether the model is available in the software, and a function 'create' that returns model object (these are as before, with adding a keras model), and additionally a definition of a class that implements the required methods. The return types of these methods should be the same as those returned by the corresponding Keras methods. The content of the *model_template.py* template file is copied below as an example.

```python
title = "Template_model"
include = False

def create(input_shape):
return TemplateModel(input_shape)

class TemplateModel:
def __init__(self, input_shape):
self.img_shape = input_shape
self.generator, self.discriminator = self.compile_custom_model()

def compile_custom_model(self):
# e.g.: compile generator, compile discriminator, return.
return 0, 0

def count_params(self):
# e.g. return self.generator.count_params()
# for the default models, the number of parameters that is returned is the
amount that are involved in processing, not in training. So for e.g. a GAN,
the discriminator params are not included.
return 0

def fit(self, train_x, train_y, epochs, batch_size=1, shuffle=True,
callbacks=[]):
for c in callbacks:
c.params['epochs'] = epochs

# fit model, e.g.:
for e in range(epochs):
for i in range(len(train_x) // batch_size):
# fit batch
pass

logs = {'loss': 0.0}
for c in callbacks:
c.on_batch_end(i, logs)

def predict(self, images):
# e.g.: return self.generator.predict(images)
return None

def save(self, path):
pass
```

```
def load(self, path):
pass
```

A more concrete example of the implementation of a custom model can be found in Ais/models/pix2pix.py. The pix2pix model (*Isola et al., 2017*) is also implemented in Keras, but since it internally uses two separate Keras model objects (the generator and the discriminator), some additional effort is required to make it compatible with the class format used in Ais. See: here and *Linder-Norén, 2019*.

## Importing and exporting models

### Export

After training, Ais models can be saved for future re-use or for uploading to the repository aiscryoet.org. The resulting savefiles, with extension *.scnm*, are uncompressed .tar archives that contain an number of required files:

- A .h5 file that fully describes the CNN architecture and weights. *The .h5 file is generated using the standard keras procedure model.save.*
- A .json metadata file, which contains information on the name and processing parameters used for the model.

If a tomogram was open in Ais at the moment of saving the model, two additional files are also included:

- A .tiff file, containing a single slice from that tomogram, saved so that the model performance can be validated prior to releasing it on the repository.
- A .png file, an image (downsized 512 × 512 pixels) depicting the validation slice with a segmentation overlaid on top, which is used as the thumbnail on the model repository.

Of these last two files, only the .png file is publically available on the repository. The image that will be publically visible is also displayed on the aiscryoet.org upload page, prior to actually uploading the model.

# Upload Model

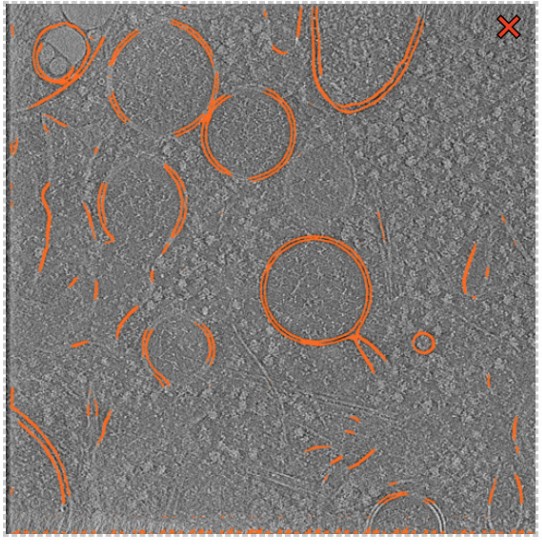

Model name:  Membrane

Model type: VGGNet

Pixel size: 14.04 Å

Box size: 64 pixels (89.86 nm)

## Additional information (optional)

Filters:　　　○ WBP　　◉ SIRT　　☐ CTF corrected

Contrast:　　◉ Dark features, light background　　○ Light features, dark background

Author name: [ Mart ]　　email: [ mgflast@gmail.com ]

Include name and email in public model information: ◉ yes　　○ no

Tags: [ Separate tags with ; ]

## References (optional)

Dataset reference: [ e.g. EMPIAR accession number ]

Article reference: [ e.g. DOI for primary reference ]

✔ I'm not a robot　　reCAPTCHA
Privacy - Terms

Submit

**Appendix 1—figure 1.** Uploading a model to the model repository via aiscryoet.org/upload.

## Import

A.scnm tar archive that contains correctly formatted.h5 and.json files can be loaded into Ais. Models generated elsewhere can also be imported, if these two files are combined into a.tar archive and renamed.scnm.

*The .h5 file is read using the standard keras load_model procedure.*
The following fields are expected in the.json:

```
{
"title": "Membrane",
"colour": [1.0, 0.40784314274787903, 0.0],
"apix": 14.04,
"compiled": true,
"box_size": 64,
"model_enum": 5,
"epochs": 50,
"batch_size": 64,
"active": true,
"blend": false,
"show": true,
"alpha": 0.75,
"threshold": 0.5,
"overlap": 0.20000000298023224,
"active_tab": 0,
"n_parameters": 17521921,
"n_copies": 10,
"info": "VGGNet L (17521921, 64, 14.040, 0.0189)",
"info_short": "(VGGNet L, 64, 14.040, 0.0189)",
"excess_negative": 100,
"emit": false,
"absorb": false,
"loss": 0.01889071986079216
}
```

## Model groups

Instead of saving single models, it is also possible to save multiple models and the interactions between them as a group.

In this case, a single file with extension.scnmgroup is created. This file is also an uncompressed.tar archive, and it contains one.scnm file for each model in the group, as well as a.json file that describes the interactions between the models.

