## [Editor Report · eLife Assessment]

This work describes a new software platform for machine-learning-based segmentation of and particle-picking in cryo-electron tomograms. The program and its corresponding online database of trained models will allow experimentalists to conveniently test different models and share their results with others. The paper provides **convincing** evidence that the software will be **valuable** to the community.

---

## [Referee Report · Reviewer #1 (Public review)]

This paper describes "Ais", a new software tool for machine-learning based segmentation and particle picking of electron tomograms. The software can visualise tomograms as slices and allows manual annotation for the training of a provided set of various types of neural networks. New networks can be added, provided they adhere to a python file with an (undescribed) format. Once networks have been trained on manually annotated tomograms, they can be used to segment new tomograms within the same software. The authors also set up an online repository to which users can upload their models, so they might be re-used by others with similar needs. By logically combining the results from different types of segmentations, they further improve the detection of distinct features. The authors demonstrate the usefulness of their software on various data sets. Thus, the software appears to be a valuable tool for the cryo-ET community that will lower the boundaries of using a variety of machine-learning methods to help interpret tomograms.

---

## [Referee Report · Reviewer #2 (Public review)]

Summary:

Last et al. present Ais, a new deep learning based software package for segmentation of cryo electron tomography data sets. The distinguishing factor of this package is its orientation to the joint use of different models, rather than the implementation of a given approach: Notably, the software is supported by an online repository of segmentation models, open to contributions from the community.

The usefulness of handling different models in one single environment is showcased with a comparative study on how different models perform on a given data set; then with an explanation on how the results of several models can be manually merged by the interactive tools inside Ais.

The manuscripts presents two applications of Ais on real data sets; one oriented to showcase its particle picking capacities on a study previously completed by the authors; a second one refers to a complex segmentation problem on two different data sets (representing different geometries as bacterial cilia and mitochondria in a mouse neuron), both from public databases.

The software described in the paper is compactly documented in its website, additionally providing links to some youtube videos (less than an hour it toral) where the authors videocapture and comment major workflows.

In short, the manuscript describes a valuable resource for the community of tomography practitioners.

Strengths:

Public repository of segmentation models; easiness of working with several models and comparing/merging the results.

---

## [Referee Report · Reviewer #3 (Public review)]

Summary:

In this manuscript, Last and colleagues describe Ais, an open-source software package for the semi-automated segmentation of cryo-electron tomography (cryo-ET) maps. Specifically, Ais provides a graphical user interface (GUI) for the manual segmentation and annotation of specific features of interest. These manual annotations are then used as input ground-truth data for training a convolutional neural network (CNN) model, which can then be used for automatic segmentation. Ais provides the option of several CNNs so that users can compare their performance on their structures of interest in order to determine the CNN that best suits their needs. Additionally, pretrained models can be uploaded and shared to an online database.

Algorithms are also provided to characterize "model interactions" which allows users to define heuristic rules on how the different segmentations interact. For instance, a membrane adjacent protein can have rules where it must colocalize a certain distance away from a membrane segmentation. Such rules can help reduce false positives; as in the case above, false negatives predicted away from membranes are eliminated.

The authors then show how Ais can be used for particle picking and subsequent subtomogram averaging and for segmentation of cellular tomograms for visual analysis. For subtomogram averaging, they used a previously published dataset and compared the averages of their automated picking with the published manual picking. Analysis of cellular tomogram segmentations were primarily visual.

Strengths:

CNN-based segmentation of cryo-ET data is a rapidly developing area of research, as it promises substantially faster results than manual segmentation as well as the possibility for higher accuracy. However, this field is still very much in the development and the overall performance of these approaches, even across different algorithms, still leaves much to be desired. In this context, I think Ais is an interesting packages, as it aims to provide both new and experienced users streamlined approaches for manual annotation, access to a number of CNNs, and methods to refine the outputs of CNN models against each other. I think this can be quite useful for users, particularly as these methods develop.

---

## [Author Response]

The following is the authors’ response to the original reviews.

We would like to thank the reviewers for helping us improve our article and software. The feedback that we received was very helpful and constructive, and we hope that the changes that we have made are indeed effective at making the software more accessible, the manuscript clearer, and the online documentation more insightful as well. A number of comments related to shared concerns, such as:

• the need to describe various processing steps more clearly (e.g. particle picking, or the nature of ‘dust’ in segmentations)

• describing the features of Ais more clearly, and explaining how it can interface with existing tools that are commonly used in cryoET

• a degree of subjectivity in the discussion of results (e.g. about Pix2pix performing better than other networks in some cases.)

We have now addressed these important points, with a focus on streamlining not only the workflow within Ais but also making interfacing between Ais and other tools easier. For instance, we explain more clearly which file types Ais uses and we have added the option to export .star files for use in, e.g., Relion, or meshes instead of coordinate lists. We also include information in the manuscript about how the particle picking process is implemented, and how false positives (‘dust’) can be avoided. Finally, all reviewers commented on our notion that Pix2pix can work ‘better’ despite reaching a higher loss after training. As suggested, we included a brief discussion about this idea in the supplementary information (Fig. S6) and used it to illustrate how Ais enables iteratively improving segmentation results.

Since receiving the reviews we have also made a number of other changes to the software that are not discussed below but that we nonetheless hope have made the software more reliable and easier to use. These include expanding the available settings, slight changes to the image processing that can help speed it up or avoid artefacts in some cases, improving the GUI-free usability of Ais, and incorporating various tools that should help make it easier to use Ais with remote data (e.g. doing annotation on an office PC, but model training on a more powerful remote PC). We have also been in contact with a number of users of the software, who reported issues or suggested various other miscellaneous improvements, and many of whom had found the software via the reviewed preprint.

**Reviewer 1 (Public Review):**
This paper describes "Ais", a new software tool for machine-learning-based segmentation and particle picking of electron tomograms. The software can visualise tomograms as slices and allows manual annotation for the training of a provided set of various types of neural networks. New networks can be added, provided they adhere to a Python file with an (undescribed) format. Once networks have been trained on manually annotated tomograms, they can be used to segment new tomograms within the same software. The authors also set up an online repository to which users can upload their models, so they might be re-used by others with similar needs. By logically combining the results from different types of segmentations, they further improve the detection of distinct features. The authors demonstrate the usefulness of their software on various data sets. Thus, the software appears to be a valuable tool for the cryo-ET community that will lower the boundaries of using a variety of machine-learning methods to help interpret tomograms.

We thank the reviewer for their kind feedback and for taking the time to review our article. On the basis of their comments, we have made a number of changes to the software, article, and documentation, that we think have helped improve the project and render it more accessible (especially for interfacing with different tools, e.g. the suggestions to describe the file formats in more detail). We respond to all individual comments one-by-one below.

**Recommendations:**
I would consider raising the level of evidence that this program is useful to *convincing* if the authors would adequately address the suggestions for improvement below.(1) It would be helpful to describe the format of the Python files that are used to import networks, possibly in a supplement to the paper.

We have now included this information in both the online documentation and as a supplementary note (Supplementary Note 1).

(2) Likewise, it would be helpful to describe the format in which particle coordinates are produced. How can they be used in subsequent sub-tomogram averaging pipelines? Are segmentations saved as MRC volumes? Or could they be saved as triangulations as well? More implementation details like this would be good to have in the paper, so readers don't have to go into the code to investigate.

Coordinates: previously, we only exported arrays of coordinates as tab-separated .txt files, compatible with e.g. EMAN2. We now added a selection menu where users can specify whether to export either .star files or tsv .txt files, which together we think should cover most software suites for subtomogram averaging.

Triangulations: We have now improved the functionality for exporting triangulations. In the particle picking menu, there is now the option to output either coordinates or meshes (as .obj files). This was previously possible in the Rendering tab, but with the inclusion in the picking menu exporting triangulations can now be done for all tomograms at once rather than manually one by one.

Edits in the text: the output formats were previously not clear in the text. We have now included this information in the introduction:

“[…] To ensure compatibility with other popular cryoET data processing suites, Ais employs file formats that are common in the field, using .mrc files for volumes, tab-separated .txt or .star files for particle datasets, and the .obj file format for exporting 3D meshes.”

(3) In Table 2, pix2pix has much higher losses than alternatives, yet the text states it achieves fewer false negatives and fewer false positives. An explanation is needed as to why that is. Also, it is mentioned that a higher number of epochs may have improved the results. Then why wasn't this attempted?

The architecture of Pix2pix is quite different from that of the other networks included in the test. Whereas all others are trained to minimize a binary cross entropy (BCE) loss, Pix2pix uses a composite loss function that is a weighted combination of the generator loss and a discriminator penalty, neither of which employ BCE. However, to be able to compare loss values, we do compute a BCE loss value for the Pix2pix generator after every training epoch. This is the value reported in the manuscript and in the software. Although Pix2pix’ BCE loss does indeed diminish during training, the model is not actually optimized to minimize this particular value and a comparison by BCE loss is therefore not entirely fair to Pix2pix. This is pointed out (in brief) in the legend to the able:

“Unlike the other architectures, Pix2pix is not trained to minimize the bce loss but uses a different loss function instead. The bce loss values shown here were computed after training and may not be entirely comparable.”

Regarding the extra number of epochs for Pix2pix: here, we initially ran in to the problem that the number of samples in the training data was low for the number of parameters in Pix2pix, leading to divergence later during training. This problem did not occur for most other models, so we decided to keep the data for the discussion around Table 1 and Figure 2 limited to that initial training dataset. After that, we increased the sample size (from 58 to 170 positive samples) and trained the model for longer. The resulting model was used in the subsequent analyses. This was previously implicit in the text but is now mentioned explicitly and in a new supplementary figure.

“For the antibody platform, the model that would be expected to be one of the worst based on the loss values, Pix2pix, actually generates segmentations that are seem well-suited for the downstream processing tasks. It also output fewer false positive segmentations for sections of membranes than many other models, including the lowest-loss model UNet. Moreover, since Pix2pix is a relatively large network, it might also be improved further by increasing the number of training epochs. We thus decided to use Pix2pix for the segmentation of antibody platforms, and increased the size of the antibody platform training dataset (from 58 to 170 positive samples) to train a much improved second iteration of the network for use in the following analyses (Fig. S6).”

(4) It is not so clear what absorb and emit mean in the text about model interactions. A few explanatory sentences would be useful here.

We have expanded this paragraph to include some more detail.

“Besides these specific interactions between two models, the software also enables pitching multiple models against one another in what we call ‘model competition’. Models can be set to ‘emit’ and/or ‘absorb’ competition from other models. Here, to emit competition means that a model’s prediction value is included in a list of competing models. To absorb competition means that a model’s prediction value will be compared to all values in that list, and that this model’s prediction value for any pixel will be set to zero if any of the competing models’ prediction value is higher. On a pixel-by-pixel basis, all models that absorb competition are thus suppressed whenever their prediction value for a pixel is lower than that of any of the emitting models.”

(5) Under Figure 4, the main text states "the model interactions described above", but because multiple interactions were described it is not clear which ones they were. Better to just specify again.

Changed as follows:

“The antibody platform and antibody-C1 complex models were then applied to the respective datasets, in combination with the membrane and carbon models and the model interactions described above (Fig. 4b): the membrane avoiding carbon, and the antibody platforms colocalizing with the resulting membranes”.

(6) The next paragraph mentions a "batch particle picking process to determine lists of particle coordinates", but the algorithm for how coordinates are obtained from segmented volumes is not described.

We have added a paragraph to the main text to describe the picking process:

“This picking step comprises a number of processing steps (Fig. S7). First, the segmented (.mrc) volumes are thresholded at a user-specified level. Second, a distance transform of the resulting binary volume is computed, in which every nonzero pixel in the binary volume is assigned a new value, equal to the distance of that pixel to the nearest zero-valued pixel in the mask. Third, a watershed transform is applied to the resulting volume, so that the sets of pixels closest to any local maximum in the distance transformed volume are assigned to one group. Fourth, groups that are smaller than a user-specified minimum volume are discarded. Fifth, groups are assigned a weight value, equal to the sum of the prediction value (i.e. the corresponding pixel value in the input .mrc volume) of the pixels in the group. For every group found within close proximity to another group (using a user-specified value for the minimum particle spacing), the group with the lower weight value is discarded. Finally, the centroid coordinate of the grouped pixels is considered the final particle coordinate, and the list of all coordinates is saved in a tab-separated text file.

“As an alternative output format, segmentations can also be converted to and saved as triangulated meshes, which can then be used for, e.g., membrane-guided particle picking. After picking particles, the resulting coordinates are immediately available for inspection in the Ais 3D renderer (Fig. S8).“

The two supplementary figures are pasted below for convenience. Fig. S7 is new, while Fig. S8 was previously Fig. S10 -the reference to this figure was originally missing in the main text, but is now included.

(7) In the Methods section, it is stated that no validation splits are used "in order to make full use of an input set". This sounds like an odd decision, given the importance of validation sets in the training of many neural networks. Then how is overfitting monitored or prevented? This sounds like a major limitation of the method.

In our experience, the best way of preparing a suitable model is to (iteratively) annotate a set of training images and visually inspect the result. Since the manual annotation step is the bottleneck in this process, we decided not to use validation split in order to make full use of an annotated training dataset (i.e. a validation split of 20% would mean that 20% of the manually annotated training data is not used for training)

We do recognize the importance of using separate data for validation, or at least offering the possibility of doing so. We have now added a parameter to the settings (and made a Settings menu item available in the top menu bar) where users can specify what fraction (0, 10, 20, or 50%) of training datasets should be set aside for validation. If the chosen value is not 0%, the software reports the validation loss as well as the size of the split during training, rather than (as was done previously) the training loss. We have, however, set the default value for the validation split to 0%, for the same reason as before. We also added a section to the online documentation about using validation splits, and edited the corresponding paragraph in the methods section:

“The reported loss is that calculated on the training dataset itself, i.e., no validation split was applied. During regular use of the software, users can specify whether to use a validation split or not. By default, a validation split is not applied, in order to make full use of an input set of ground truth annotations. Depending on the chosen split size, the software reports either the overall training loss or the validation loss during training.”

(8) Related to this point: how is the training of the models in the software modelled? It might be helpful to add a paragraph to the paper in which this process is described, together with indicators of what to look out for when training a model, e.g. when should one stop training?

We have expanded the paragraph where we write about the utility of comparing different networks architectures to also include a note on how Ais facilitates monitoring the output of a model during training:

“When taking the training and processing speeds in to account as well as the segmentation results, there is no overall best architecture. We therefore included multiple well-performing model architectures in the final library, in order to allow users to select from these models to find one that works well for their specific datasets. Although it is not necessary to screen different network architectures and users may simply opt to use the default (VGGNet), these results thus show that it can be useful to test different networks in order to identify one that is best. Moreover, these results also highlight the utility of preparing well-performing models by iteratively improving training datasets and re-training models in a streamlined interface. To aid in this process, the software displays the loss value of a network during training and allows for the application of models to datasets during training. Thus, users can inspect how a model’s output changes during training and decide whether to interrupt training and improve the training data or choose a different architecture.”

(9) Figure 1 legend: define the colours of the different segmentations.

Done

(10) It may be better to colour Figure 2B with the same colours as Figure 2A.

We tried this, but the effect is that the underlying density is much harder to see. We think the current grayscale image paired with the various segmentations underneath is better for visually identifying which density corresponds to membranes, carbon film, or antibody platforms.

**Reviewer 2 (Public Review):**
Summary:Last et al. present Ais, a new deep learning-based software package for the segmentation of cryo-electron tomography data sets. The distinguishing factor of this package is its orientation to the joint use of different models, rather than the implementation of a given approach. Notably, the software is supported by an online repository of segmentation models, open to contributions from the community.The usefulness of handling different models in one single environment is showcased with a comparative study on how different models perform on a given data set; then with an explanation of how the results of several models can be manually merged by the interactive tools inside Ais.The manuscripts present two applications of Ais on real data sets; one is oriented to showcase its particlepicking capacities on a study previously completed by the authors; the second one refers to a complex segmentation problem on two different data sets (representing different geometries as bacterial cilia and mitochondria in a mouse neuron), both from public databases.The software described in the paper is compactly documented on its website, additionally providing links to some YouTube videos (less than an hour in total) where the authors videocapture and comment on major workflows.In short, the manuscript describes a valuable resource for the community of tomography practitioners.Strengths:A public repository of segmentation models; easiness of working with several models and comparing/merging the results.Weaknesses:A certain lack of concretion when describing the overall features of the software that differentiate it from others.

We thank the reviewer for their kind and constructive feedback. Following the suggestion to use the Pix2pix results to illustrate the utility of Ais for analyzing results, we have added a new supplementary figure (Fig. S6) and brief discussion, showing the use of Ais in iteratively improving segmentation results. We have also expanded the online documentation and included a note in the supplementary information about how models are saved/loaded (Supplemetary note 1).

**Recommendations:**
I would like to ask the authors about some concerns about the Ais project as a whole:(1) The website that accompanies the paper (aiscryoet.org), albeit functional, seems to be in its first steps. Is it planned to extend it? In particular, one of the major contributions of the paper (the maintenance of an open repository of models) could use better documentation describing the expected formats to submit models. This could even be discussed in the supplementary material of the manuscript, as this feature is possibly the most distinctive one of the paper. Engaging third-party users would require giving them an easier entry point, and the superficial mention of this aspect in the online documentation could be much more generous.

We have added a new page to the online documentation, titled ‘Sharing models’ where we include an explanation of the structure of model files and demonstrate the upload page. We also added a note to the Supplementary Information that explains the file format for models, and how they are loaded/saved (i.e., that these standard keras model obects).

To make it easier to interface Ais with other tools, we have now also made some of the core functionality available (e.g. training models, batch segmentation) via the command line interface. Information on how to use this is included in the online documentation. All file formats are common formats used in cryoET, so that using Ais in a workflow with, e.g. AreTomo -> Ais -> Relion should now be more straightforward.

(2) A different major line advanced by the authors to underpin the novelty of the software, is its claimed flexibility and modularity. In particular, the restrictions of other packages in terms of visualization and user interaction are mentioned. Although in the manuscript it is also mentioned that most of the functionalities in Ais are already available in major established packages, as a reader I am left confused about what exactly makes the offer of Ais different from others in terms of operation and interaction: is it just the two aspects developed in the manuscript (possibility of using different models and tools to operate model interaction)? If so, it should probably be stated; but if the authors want to pinpoint other aspects of the capacity of Ais to drive smoothly the interactions, they should be listed and described, instead of leaving it as an unspecific comment. As a potential user of Ais, I would suggest the authors add (maybe in the supplementary material) a listing of such features. Figure 1 does indeed carry the name "overview of (...) functionalities", but it is not clear to me which functionalities I can expect to be absent or differently solved on the other tools they mention.

We have rewritten the part of the introduction where we previously listed the features as below. We think it should now be clearer for the reader to know what features to expect, as well as how Ais can interface with other software (i.e. what the inputs and outputs are). We have also edited the caption for Figure 1 to make it explicit that panels A to C represent the annotation, model preparation, and rendering steps of the Ais workflow and that the images are screenshots from the software.

“In this report we present Ais, an open-source tool that is designed to enable any cryoET user – whether experienced with software and segmentation or a novice – to quickly and accurately segment their cryoET data in a streamlined and largely automated fashion. Ais comprises a comprehensive and accessible user interface within which all steps of segmentation can be performed, including: the annotation of tomograms and compiling datasets for the training of convolutional neural networks (CNNs), training and monitoring performance of CNNs for automated segmentation, 3D visualization of segmentations, and exporting particle coordinates or meshes for use in downstream processes. To help generate accurate segmentations, the software contains a library of various neural network architectures and implements a system of configurable interactions between different models. Overall, the software thus aims to enable a streamlined workflow where users can interactively test, improve, and employ CNNs for automated segmentation. To ensure compatibility with other popular cryoET data processing suites, Ais employs file formats that are common in the field, using .mrc files for volumes, tab-separated .txt or .star files for particle datasets, and the .obj file format for exporting 3D meshes.”

“Figure 1 – an overview of the user interface and functionalities. The various panels represent sequential stages in the Ais processing workflow, including annotation (a), testing CNNs (b), visualizing segmentation (c). These images (a-c) are unedited screenshots of the software. (a) […]”

(3) Table 1 could have the names of the three last columns. The table has enough empty space in the other columns to accommodate this.

Done.

(4) The comment about Pix2pix needing a larger number of training epochs (being a larger model than the other ones considered) is interesting. It also lends itself for the authors to illustrate the ability of their software to precisely do this: allow the users to flexibly analyze results and test hypothesis

Please see the response to Reviewer 1 comment #3. We agree that this is a useful example of the ability to iterate between annotation and training, and have added an explicit mention of this in the text:

“Moreover, since Pix2pix is a relatively large network, it might also be improved further by increasing the number of training epochs. In a second iteration of annotation and training, we thus increased the size of the antibody platform training dataset (from 58 to 170 positive samples) and generated an improved Pix2pix model for use in the following analyses.”

**Reviewer 3 (Public Review):**
We appreciate the reviewer’s extensive and very helpful feedback and are glad to read that they consider Ais potentially quite useful for the users. To address the reviewer’s comments, we have made various edits to the text, figures, and documentation, that we think have helped improve the clarity of our work. We list all edits below.SummaryIn this manuscript, Last and colleagues describe Ais, an open-source software package for the semi-automated segmentation of cryo-electron tomography (cryo-ET) maps. Specifically, Ais provides a graphical user interface (GUI) for the manual segmentation and annotation of specific features of interest. These manual annotations are then used as input ground-truth data for training a convolutional neural network (CNN) model, which can then be used for automatic segmentation. Ais provides the option of several CNNs so that users can compare their performance on their structures of interest in order to determine the CNN that best suits their needs. Additionally, pre-trained models can be uploaded and shared to an online database.Algorithms are also provided to characterize "model interactions" which allows users to define heuristic rules on how the different segmentations interact. For instance, a membrane-adjacent protein can have rules where it must colocalize a certain distance away from a membrane segmentation. Such rules can help reduce false positives; as in the case above, false negatives predicted away from membranes are eliminated.The authors then show how Ais can be used for particle picking and subsequent subtomogram averaging and for the segmentation of cellular tomograms for visual analysis. For subtomogram averaging, they used a previously published dataset and compared the averages of their automated picking with the published manual picking. Analysis of cellular tomogram segmentation was primarily visual.Strengths:CNN-based segmentation of cryo-ET data is a rapidly developing area of research, as it promises substantially faster results than manual segmentation as well as the possibility for higher accuracy. However, this field is still very much in the development and the overall performance of these approaches, even across different algorithms, still leaves much to be desired. In this context, I think Ais is an interesting package, as it aims to provide both new and experienced users with streamlined approaches for manual annotation, access to a number of CNNs, and methods to refine the outputs of CNN models against each other. I think this can be quite useful for users, particularly as these methods develop.Weaknesses:Whilst overall I am enthusiastic about this manuscript, I still have a number of comments:(1) On page 5, paragraph 1, there is a discussion on human judgement of these results. I think a more detailed discussion is required here, as from looking at the figures, I don't know that I agree with the authors' statement that Pix2pix is better. I acknowledge that this is extremely subjective, which is the problem. I think that a manual segmentation should also be shown in a figure so that the reader has a better way to gauge the performance of the automated segmentation.

Please see the answer to Reviewer 1’s comment #3.

(2) On page 7, the authors mention terms such as "emit" and "absorb" but never properly define them, such that I feel like I'm guessing at their meaning. Precise definitions of these terms should be provided.

We have expanded this paragraph to include some more detail:

“Besides these specific interactions between two models, the software also enables pitching multiple models against one another in what we call ‘model competition’. Models can be set to ‘emit’ and/or ‘absorb’ competition from other models. Here, to emit competition means that a model’s prediction value is included in a list of competing models. To absorb competition means that a model’s prediction value will be compared to all values in that list, and that this model’s prediction value for any pixel will be set to zero if any of the competing models’ prediction value is higher. On a pixel-by-pixel basis, all models that absorb competition are thus suppressed whenever their prediction value for a pixel is lower than that of any of the emitting models.”

(3) For Figure 3, it's unclear if the parent models shown (particularly the carbon model) are binary or not.The figure looks to be grey values, which would imply that it's the visualization of some prediction score. If so, how is this thresholded? This can also be made clearer in the text.

The figures show the grayscale output of the parent model, but this grayscale output is thresholded to produce a binary mask that is used in an interaction. We have edited the text to include a mention of thresholding at a user-specified threshold value:

“These interactions are implemented as follows: first, a binary mask is generated by thresholding the parent model’s predictions using a user-specified threshold value. Next, the mask is then dilated using a circular kernel with a radius 𝑅, a parameter that we call the interaction radius. Finally, the child model’s prediction values are multiplied with this mask.”

To avoid confusion, we have also edited the figure to show the binary masks rather than the grayscale segmentations.

(4) Figure 3D was produced in ChimeraX using the hide dust function. I think some discussion on the nature of this "dust" is in order, e.g. how much is there and how large does it need to be to be considered dust? Given that these segmentations can be used for particle picking, this seems like it may be a major contributor to false positives.

‘Dust’ in segmentations is essentially unavoidable; it would require a perfect model that does not produce any false positives. However, when models are sufficiently accurate, the volume of false positives is typically smaller than that of the structures that were intended to be segmented. In these cases, discarding particles based on size is a practical way of filtering the segmentation results. Since it is difficult to generalize when to consider something ‘dust’ we decided to include this additional text in the Method’s section rather than in the main text:

“… with the use of the ‘hide dust’ function (the same settings were used for each panel, different settings used for each feature).

This ‘dust’ corresponds to small (in comparison to the segmented structures of interest) volumes of false positive segmentations, which are present in the data due to imperfections in the used models. The rate and volume of false positives can be reduced either by improving the models (typically by including more examples of the images of what would be false negatives or positives in the training data) or, if the dust particles are indeed smaller than the structures of interest, they can simply be discarded by filtering particles based on their volume, as applied here. In particle picking a ‘minimum particle volume’ is specified – particles with a smaller volume are considered ‘dust’.

In combination with the newly included text about the method of converting volumes into lists of coordinates (see Reviewer 1’s comment #6).

“Third, a watershed transform is applied to the resulting volume, so that the sets of pixels closest to any local maximum in the distance transformed volume are assigned to one group. Fourth, groups that are smaller than a user-specified minimum volume are discarded…”

We think it should now be clearer that (some form of) discarding ‘dust’ is a step that is typically included in the particle picking process.

(5) Page 9 contains the following sentence: "After selecting these values, we then launched a batch particle picking process to determine lists of particle coordinates based on the segmented volumes." Given how important this is, I feel like this requires significant description, e.g. how are densities thresholded, how are centers determined, and what if there are overlapping segmentations?

Please see the response to Reviewer 1’s comment #6.

(6) The FSC shown in Figure S6 for the auto-picked maps is concerning. First, a horizontal line at FSC = 0 should be added. It seems that starting at a frequency of ~0.045, the FSC of the autopicked map increases above zero and stays there. Since this is not present in the FSC of the manually picked averages, this suggests the automatic approach is also finding some sort of consistent features. This needs to be discussed.

Thank you for pointing this out. Awkwardly, this was due to a mistake made while formatting the figure. In the two separate original plots, the Y axes had slightly different ranges, but this was missed when they were combined to prepare the joint supplementary figure. As a result, the FSC values for the autopicked half maps are displayed incorrectly. The original separate plots are shown below to illustrate the discrepancy:

**Author response image 1. sa4fig1:** 

The corrected figure is Figure S9 in the manuscript. The values of 44 Å and 46 Å were not determined from the graph and remain unchanged.

(7) Page 11 contains the statement "the segmented volumes found no immediately apparent false positive predictions of these pores". This is quite subjective and I don't know that I agree with this assessment. Unless the authors decide to quantify this through subtomogram classification, I don't think this statement is appropriate.

We originally included this statement and the supplementary figure because we wanted to show another example of automated picking, this time in the more crowded environment of the cell. We do agree that it requires better substantiation, but also think that the demonstration of automated picking of the antibody platforms and IgG3-C1 complexes for subtomogram averaging suffices to demonstrate Ais’ picking capabilities. Since the supplementary information includes an example of picked coordinates rendered in the Ais 3D viewer (Figure S7) that also used the pore dataset, we still include the supplementary figure (S10) but have edited the statement to read:

“Moreover, we could identify the molecular pores within the DMV, and pick sets of particles that might be suitable for use in subtomogram averaging (see Fig. S11).”

We have also expanded the text that accompanies the supplementary figure to emphasize that results from automated picking are likely to require further curation, e.g. by classification in subtomogram averaging, and that the selection of particles is highly dependent on the thresholds used in the conversion from volumes to lists of coordinates.

(8) In the methods, the authors note that particle picking is explained in detail in the online documentation. Given that this is a key feature of this software, such an explanation should be in the manuscript.

Please see the response to Reviewer 1’s comment #6.

**Recommendations:**
(9) The word "model" seems to be used quite ambiguously. Sometimes it seems to refer to the manual segmentations, the CNN architectures, the trained models, or the output predictions. More precision in this language would greatly improve the readability of the manuscript.

This was indeed quite ambiguous, especially in the introduction. We have edited the text to be clearer on these differences. The word ‘model’ is now only used to refer to trained CNNs that segment a particular feature (as in ‘membrane model’ or ‘model interactions’). Where we used terms such as ‘3D models’ to describe scenes rendered in 3D, we now use ‘3D visualizations’ or similar terms. Where we previously used the term ‘models’ to refer to CNN architectures, we now use terms such as ‘neural network architectures’ or ‘architecture’. Some examples:

… with which one can automatically segment the same or any other dataset …

Moreover, since Pix2pix is a relatively large network, …

… to generate a 3D visualization of ten distinct cellular …

… with the use of the same training datasets for all network architectures …

In Figure 1, the text in panels D and E is illegible.

We have edited the figure to show the text more clearly (the previous images were unedited screenshots of the website).

(10) Prior to the section on model interactions, I was under the impression that all annotations were performed simultaneously. I think it could be clarified that models are generated per annotation type.

Multiple different features can be annotated (i.e. drawn by hand by the user) at the same time, but each trained CNN only segments one feature. CNNs that output segmentations for multiple features can be implemented straightforwardly, but this introduces the need to provide training data where for every grayscale image, every feature is annotated. This can make preparing the training data much more cumbersome. Reusability of the models is also hampered. We now mention the separateness of the networks explicitly in the introduction:

“Multiple features, such as membranes, microtubules, ribosomes, and phosphate crystals, can be segmented and edited at the same time across multiple datasets (even hundreds). These annotations are then extracted and used as ground truth labels upon which to condition multiple separate neural networks, …”

(11) On page 6, there is the text "some features are assigned a high segmentation value by multiple of the networks, leading to ambiguity in the results". Do they mean some false features?

To avoid ambiguity of the word ‘features’, we have edited the sentence to read:

“… some parts of the image are assigned a high segmentation value by multiple of the networks, leading to false classifications and ambiguity in the results.”

(12) Figures 2 and 3 would be easier to follow if they had consistent coloring.

We have changed the colouring in Figure 2 to match that of Figure 3 better:

(13) For Figure 3D, I'm confused as to why the authors showed results from the tomogram in Figure 2B. It seems like the tomogram in Figure 3C would be a more obvious choice, as we would be able to see how the 2D slices look in 3D. This would also make it easier to see the effect of interactions on false negatives. Also, since the orientation of the tomogram in 2B is quite different than that shown in 3D, it's a bit difficult to relate the two.

We chose to show this dataset because it exemplifies the effects of both model competition and model interactions better than the tomogram in Figure 3C. See Figure 3D and Author response image 2 for a comparison:

**Author response image 2. sa4fig2:** 

(14) I'm confused as to why the tomographic data shown in Figures 4D, E, and F are black on white while all other cryo-ET data is shown as white on black.

The images in Figure 4DEF are now inverted.

(15) For Figure 5, there needs to be better visual cueing to emphasize which tomographic slices are related to the segmentations in Panels A and B.

We have edited the figure to show more clearly which grayscale image corresponds to which segmentation:

(16) I don't understand what I should be taking away from Figures S1 and S2. There are a lot of boxes around membrane areas and I don't know what these boxes mean.

We have added a more descriptive text to these figures. The boxes are placed by the user to select areas of the image that will be sampled when saving training datasets.